# MODEL CHANGELISTS: CHARACTERIZING CHANGES IN ML PREDICTION APIS

## ABSTRACT

Updates to Machine Learning as a Service (MLaaS) APIs may affect downstream systems that depend on their predictions. However, performance changes introduced by these updates are poorly documented by providers and seldom studied in the literature. As a result, users are left wondering: *do model updates introduce subtle performance changes that could adversely affect my system?* Ideally, users would have access to a detailed `ChangeList` specifying the slices of data where model performance has improved and degraded since the update. But, producing a `ChangeList` is challenging because it requires (1) discovering slices in the absence of detailed annotations or metadata, (2) accurately attributing coherent concepts to the discovered slices, and (3) communicating them to the user in a digestable manner. We introduce `Mocha`, an interactive framework for building, verifying and releasing `ChangeLists` that addresses these challenges. Using it, we perform a large-scale analysis of three real-world MLaaS API updates. We produce a `ChangeList` for each, identifying over 100 coherent data slices on which the model's performance changed significantly. Notably, we find 63 instances where an update improves performance globally, but hurts performance on a coherent slice – a phenomenon not previously documented at scale in the literature. These findings underscore the importance of producing a detailed `ChangeList` when the model behind an API is updated.

## 1 INTRODUCTION

Modern software systems often depend on Machine Learning as a Service (MLaaS) APIs developed by cloud providers (*e.g.* AWS, GCP, Azure) or research organizations (*e.g.* OpenAI, HuggingFace). The models behind these APIs are periodically updated and new versions are released. However, to the user, how a new update will affect the workings of their broader system is typically unclear. Consider, for example, a newspaper that uses an image tagging API to source archival photos for retrospective stories (Greenfield, 2018). Updates to the underlying model could lead to unexpected changes in the workflow of photo editors and journalists who rely on the system.

MLaaS providers rarely provide transparent evaluations of their updates, and those that do focus on global metrics and vague notions of improvement. Release notes from cloud providers like Amazon, Google and Microsoft for their APIs are terse and provide little information e.g. Microsoft's Vision API (Feb '22 update) noting "general performance and AI quality improvements" (Microsoft, b).

These release notes tell an incomplete story: saying that one model improves on another obscures the fact that models may perform very differently on fine-grained slices of data (Ribeiro et al., 2020; de Vries et al., 2019). Returning to the newspaper described above, performance after the update may improve globally, while still deteriorating on historic photos – the kind of photos commonly found in the newspaper's archives. Without more detailed evaluations, users are left wondering:

> *Do model updates introduce subtle changes that could adversely affect my system?*

While many studies include detailed comparisons of MLaaS APIs (Buolamwini & Gebru, 2018; Goel et al., 2021a;b; Ribeiro et al., 2020; Qi et al., 2020), they lack comparisons of the *same* API before and after an update. Recent work shows that API updates can lead to performance drops on benchmarks (Chen et al., 2021), but the analysis is limited to simple tasks and global measurements.

Figure 1: **Overview of Mocha.** *(left)* An MLaaS API updates and changes predictions for downstream users; *(right)* Mocha is an interactive framework for building, verifying and releasing ChangeLists to explain model updates using slices of data.

Answering this question would be easier if providers released detailed reports specifying the slices of data where performance has changed. We formalize this using the notion of a ChangeList. Ideally, a ChangeList is interactive, allowing a user to explore how the model's behavior has changed on the slices most important to their system. For the example above, the newspaper should be able to draw conclusions like: "the updated API detects objects in historic photos with 10% lower recall". Such conclusions would inform decisions around whether or not to integrate the update. However, producing a comprehensive ChangeList is difficult due to 3 main challenges:

1. For complex data like images, the set of slices that partition the data is extremely large and unknown a priori. *How can we gather coherent slices that explain the change?*

2. When interpreting slices, we typically attribute concepts (*e.g. historic*) to them. However, if the slice was discovered automatically, it may not align perfectly with a concept, leading to false conclusions about performance on the concept. *How do we quickly perform accurate attribution?*

3. The number of slices with significant changes can be very large, and not all changes will be relevant to all users. *How do we help users surface slices most important to their system?*

To address these challenges we introduce Mocha, an interactive framework for building, verifying and releasing ChangeLists for model updates. Mocha consists of three phases:

1. **Discovery**: First, we adapt a recently proposed slice discovery method (Eyuboglu et al., 2022) to gather slices for the ChangeList in Mocha. We use cross-modal embeddings and a simple mixture model to identify slices of data where the models differ. Mocha also supports manual slicing over metadata, and can incorporate slices from any method of slice discovery.

2. **Attribution**: Next, we ascribe concepts to the discovered slices. Via an interactive process termed *micro-labeling*, we verify the accuracy of the attributions and dynamically correct them. Cross-modal embeddings (*e.g.* CLIP) are used to guide an importance sampler (Owen, 2013) that surfaces a small number of examples for labeling. Labeled examples are then used to estimate the precision and recall of the user attributions, and to update slices to reflect label feedback.

3. **Release**: Finally, to help users understand model updates, we release the ChangeList in the Mocha web interface. The slices in the ChangeList are indexed by cross-modal embeddings, and are therefore easily searchable by text or image. Further, if the ChangeList is missing slices important to the user, they can initiate discovery and attribution to edit the ChangeList.

While Mocha can be used to prepare ChangeLists for any pair of models, we focus particularly on demonstrating its application to the challenging real-world problem of documenting MLaaS APIs. We use Mocha to study updates to three image tagging APIs with the HAPI database (Chen et al.), which gathers predictions on the same test examples before and after an update. We produce one ChangeList per API update, with findings from our study of ChangeLists below:

- The ChangeLists include over 100 coherent slices on which the model's performance changed significantly. These slices were not annotated in the dataset and were discovered by Mocha.

- There are 63 slices in the ChangeLists on which an API update introduced a statistically significant degradation in performance. For example, between 2020 and 2022, the accuracy of a model from Google Cloud Vision on the task of tagging "people" degraded by 17.7%-points for

black and white images. This phenomenon, where an update improves performance globally but hurts performance on a coherent slice, has not been documented at this scale in the literature.

Our findings underscore the importance of releasing `ChangeLists`, and we recommend that providers and research organizations release detailed `ChangeLists` alongside model updates to help users make informed decisions around integrating updates into downstream systems.

### 1.1 PRIOR WORK

**Evaluating MLaaS APIs.** A growing number of publications include evaluations of MLaaS APIs. Some evaluate a single API in depth (Hosseini et al., 2017). Others compare several different APIs on the same task (Yao et al., 2017; Reis et al., 2018; Hosseini et al., 2019). For example, Chen *et al.* compare APIs from different providers and demonstrate that performance varies by class (Chen et al., 2020). Several studies discuss significant racial disparities in the performance of MLaaS APIs (Buolamwini & Gebru, 2018; Koenecke et al., 2020). More generally, evaluation frameworks like Checklist and RobustnessGym applied to MLaaS APIs (Ribeiro et al., 2020; Goel et al., 2021b) demonstrate an array of vulnerabilities not discernible with standard evaluations. While some of these studies compare APIs from different providers, few compare different versions of an API from the *same* API. Recently, Chen et al. (2021) showed that the accuracy of ML APIs sometimes changes after an update. This analysis, which is most similar to our own, is limited to simple classification tasks and does not consider error consistency or slice-level differences in performance.

**Comparing Machine Learning Models.** Prior studies have compared machine learning models by measuring the consistency of the errors made by different image classifiers (Geirhos et al., 2020; 2021; Gontijo-Lopes et al.; Mania et al., 2019; Fort et al., 2019). For example, Mania et al. (2019) measure the consistency of errors made by different ImageNet classifiers with the same accuracy, showing that error consistency is significantly higher than would be expected if predictions from different models were independent. Building on this, recent work explores how differences in model initialization and architecture affect the consistency of errors (Gontijo-Lopes et al.; Fort et al., 2019). Instead of using a fixed set of test inputs, others generate new inputs where models disagree (Li et al., 2021; Xie et al., 2019; Pei et al., 2017) or compare outputs of explanation methods (Jia et al., 2021).

## 2 MEASURING GLOBAL CHANGES IN REAL API UPDATES

We first introduce *change metrics*, summary statistics that describe the effect of a model update on performance. Our metrics measure (1) the *performance shift* due to the update and (2) the *inconsistency* of this shift across the data. We use these metrics to provide a new perspective in the analysis of three real API updates, which motivates our proposal to build `ChangeLists` in Section 3.

**Preliminaries.** Consider a supervised learning setup where each *example* $(X, Y)$ is composed of an *input* $X \in \mathcal{X}$ (*e.g.* an image) and a *target* $Y \in \mathcal{Y}$ (*e.g.* a binary label). Assume we have a loss function (or point-wise metric) $\ell : \mathcal{Y} \times \mathcal{Y} \to \mathbb{R}$. Additionally, we have black-box access to two models trained for this task $v^{[1]}, v^{[2]} : \mathcal{X} \to \mathcal{Y}$ – e.g. these models could serve predictions for the *same* MLaaS API at different points in time: $v^{[1]}$ before an update and $v^{[2]}$ after. To compare the models, we collect their predictions $\hat{y}_i^{[j]} := v^{[j]}(x_i)$ on a dataset $\mathcal{D} = \{(x_i, y_i)\}_{i=1}^n \sim \mathcal{P}(X, Y)$.

**Change Metrics.** We define two change metrics in terms of $D = \ell(v^{[1]}(X), Y) - \ell(v^{[2]}(X), Y)$, the difference in loss between the models,

1. **Performance Shift** (*on average, did the model improve as a result of the update?*). This metric estimates the expected difference in losses,

$$\mu_\ell = \mathbb{E}[\ell(v^{[1]}(X), Y) - \ell(v^{[2]}(X), Y)] = \mathbb{E}[D]. \tag{1}$$

   Positive values of $\mu$ indicate that the model improved after the update.

2. **Performance Inconsistency** (*is the shift in performance inconsistent across the dataset?*). This metric estimates the standard deviation of the difference between the losses,

$$\sigma_\ell^2 = \text{Var}[\ell(v^{[1]}(X), Y) - \ell(v^{[2]}(X), Y)] = \text{Var}[D]. \tag{2}$$

   Larger values of $\sigma_\ell$ indicate that the models frequently disagree, and the shift is not consistent. This metric is inspired by prior consistency metrics (Geirhos et al., 2020; Gontijo-Lopes et al.).

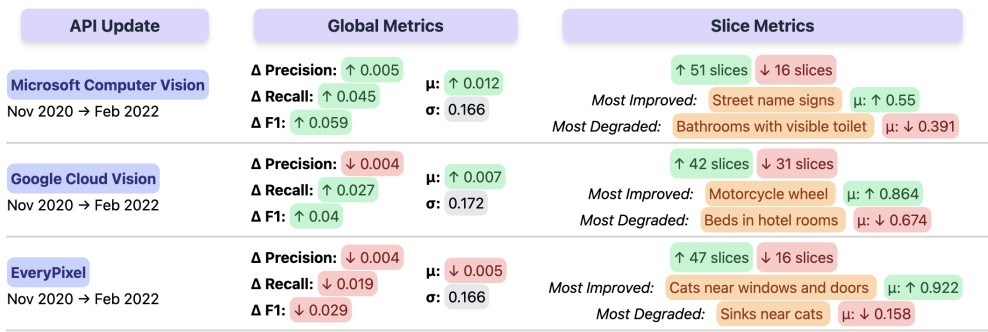

Figure 2: **Overview of Updates**. We study three real-world API updates: Microsoft (a) Computer Vision (**top**), Google Cloud Vision (**middle**), and EveryPixel Image Recognition (**bottom**). We provide background information on the update (**left**), global metrics computed on the entire dataset (**middle**, see Section 2), and a summary of the slices discovered by Mocha (**right**, see Section 4).

*Example: zero-one loss.* For the special case of the zero-one loss, $\ell_{01} : \mathcal{Y} \times \mathcal{Y} \to \{0, 1\}$, we have

$$\mu_{01} = \frac{1}{n} \sum_{i=1}^{n} ([\hat{y}^{[1]} = y_i] - [\hat{y}^{[2]} = y_i]), \qquad \sigma_{01}^2 = \frac{1}{n} \sum_{i=1}^{n} [\hat{y}^{[2]} \neq \hat{y}^{[1]}] - \mu_{01}^2.$$

Observe that $\mu_{01}$ is simply the difference in accuracy between the models, while $\sigma_{01}^2$ measures the disagreement between the models that is left over after accounting for some of the performance shift. The maximum $\sigma_{01} = 1$ occurs when both models have the same accuracy but disagree everywhere.

*Discussion.* These metrics allow us to measure when users should be cautious in using an updated model. With positive performance shift and no inconsistency, model updates can be integrated by users safely. However, high shift inconsistency even absent performance shift is concerning, since the update may disproportionately hurt performance on data important to the user's application.

**A Global Analysis of APIs.** We analyze global performance changes across three updates to real MLaaS APIs: measuring $\ell_{01}$ change metrics alongside changes in recall, precision, and F1-score. We briefly discuss the task, data, and APIs next (full details in Appendix A.4).

*Task (Image Tagging).* In image tagging, the input $X$ is an image and category (*e.g. horse*) pair and the target $Y \in \{0, 1\}$ is a binary label indicating whether an object of the category is in the image.

*Dataset (LVIS).* We use the Large Vocabulary Instance Segmentation (LVIS) dataset, a relabeling of the Common Objects in Context (COCO) images (Gupta et al., 2019; Lin et al., 2014) that reflects the breadth of categories output by image tagging APIs ($n = 1{,}577{,}603$ across 1,203 categories).

*APIs.* We consider three object detection APIs: *Microsoft* Computer Vision API, *Google* Cloud Vision API, and *EveryPixel* Image Keywording Service. The predictions are sourced from History of APIs (Chen et al.), a longitudinal database of API predictions. We additionally process the API outputs to map to labels in LVIS (details in Appendix A.4).

*Results.* From Nov '20 to Feb '22, *Google* and *Microsoft* saw accuracy shifts $\mu_{01}$ of $+0.7\%$ and $+1.2\%$ respectively ($+4.0\%$ and $+5.9\%$ F1), while *EveryPixel* saw a small degradation in $\mu_{01}$ of $-0.5\%$ ($-2.9\%$ F1). However, these shifts tell only a partial story: all three updates exhibit non-zero shift inconsistency ($\sigma_{01} > 0.15$). To put this into context, the predictions of the Google API ($\sigma_{01} = 0.172$ and $\sigma_{01}|Y = 0.326$) changed on $10+\%$ of positive examples between versions. This highlights that the *API's behavior changes in ways unexplained by global change metrics.*

## 3 CHARACTERIZING MODEL UPDATES IN DETAIL

These findings highlight the importance of producing a more detailed, fine-grained understanding of model updates. This motivates our key proposal: the introduction of a ChangeList (Section 3.1) to explain the observed performance shift and shift inconsistency using the change in fine-grained data slices, and an interactive framework called Mocha to build ChangeLists (Section 3.2).

## 3.1 CHANGELISTS: A FINE-GRAINED CHARACTERIZATION OF MODEL UPDATES

For a user, the decision on whether to use an updated model requires understanding the data examples that account for performance shift and inconsistency. Users seek explanations that focus on the data important to their application. We formalize this relationship through *slices* of data important to users, and define ChangeLists in terms of these slices in order to reflect user needs.

**Slices (S).** A *slice* is a subset of data examples that share something in common e.g. in object recognition, the set of images with dim lighting constitutes a slice. Formally, we represent a slice with a random variable $S \in \{0, 1\}$ and a set of slices with $\mathbf{S} = \{S^{(j)}\}_{j=1}^k \in \{0, 1\}^k$, with joint distribution $P(X, Y, \mathbf{S})$ over inputs, targets and slices. Each example has a realization of the slice random variables $\{s_i^{(j)}\}_{i=1}^n$. If $s_i^{(j)} = 1$, then example $(x_i, y_i)$ is in slice $S^{(j)}$. In practice, datasets do not include realizations for all possible slices, e.g. not including annotations for dim lighting. Intuitively, we would like a ChangeList to present users with slices alongside human-readable descriptions and metrics quantifying how their performance has changed. We define these next.

**Slice Attributions (A).** Define random variable $A^{(j)} \in \{0, 1\}$ to represent a text attribution for slice $S^{(j)}$ (e.g. *"dim lighting"*), with attributions $\mathbf{A} = \{A^{(j)}\}_{j=1}^k$ corresponding to $\mathbf{S}$ and example level attribute realizations $\{a_i^{(j)}\}_{i=1}^n$. If $a_i^{(j)} = 1$, then example $(x_i, y_i)$ satisfies attribution $A^{(j)}$. Typically, these realizations are unknown, and only the text attribution $A^{(j)}$ will be given.

**Slice Change Metrics (M).** Given a slice $S^{(j)}$, denote change metrics $\mu_\ell^{(j)}, \sigma_\ell^{(j)}$ for loss functions $\ell_1, \ldots, \ell_r$, with the set of change metrics for $\mathbf{S}$ denoted by $\mathbf{M} = \{\mu_{\ell_1}^{(j)}, \sigma_{\ell_1}^{(j)}, \ldots, \mu_{\ell_r}^{(j)}, \sigma_{\ell_r}^{(j)}\}_{j=1}^k$.

We are now ready to define a ChangeList using these concepts.

**Definition** (ChangeList $\mathfrak{C}$). *Given dataset $\mathcal{D}$ and models $v^{[1]}, v^{[2]}$, a ChangeList is a collection of slices $\mathbf{S}$ along with their corresponding descriptions $\mathbf{A}$ and change metrics $\mathbf{M}$.*

Given this definition, we ask: what constitutes a good ChangeList? We discuss several criteria that we expect will be desired by users of ChangeLists. These desiderata are not exhaustive, and we expect more to emerge as ChangeLists are adopted into wider practice.

1. **Diversity (of S).** Different users have different slices of interest e.g. decorators may tag images of homes, while doctors analyzing patient behavior may tag hospital images. ChangeLists should contain a diversity of slices to reflect this.
2. **Coverage (of $\sigma_\ell$ with S).** The slices $\mathbf{S}$ should together explain the inconsistency $\sigma_\ell$. The explanatory power of $\mathbf{S}$ can be measured by the coefficient of determination $r^2 = 1 - \frac{1}{\sigma_\ell^2}\mathbb{E}[(D - f(\mathbf{S}))^2]$, where $f(\mathbf{s}) = a + \mathbf{b}^T\mathbf{s}$ is a function fit by performing a linear regression of $D$ on $\mathbf{S}$.
3. **Alignment (of A with S).** Attributions should align with the examples in each slice, and users should be able to read these attributions to understand the content of each slice.
4. **Relevance (of M).** Change metrics reported in the ChangeList should be chosen to be relevant to the tasks for which the models are to be used.
5. **Navigability (of S).** Users of ChangeLists should be able to search over information in the ChangeList, including global change metrics, slices, and attributions.
6. **Editability (of $\mathfrak{C}$).** Finally, users would ideally benefit from the ability to modify a released ChangeList to meet their needs e.g. by interactively adding new slices of interest.

We emphasize that it is difficult to create a perfect ChangeList since every relevant slice cannot be anticipated ahead of time. Our interactive framework Mocha is designed to help model providers systematically build ChangeLists, while remaining flexible enough to addresses these concerns.

## 3.2 MOCHA: AN INTERACTIVE FRAMEWORK FOR GENERATING CHANGELISTS

Mocha provides an interactive interface (Fig. 3) for building, releasing and reading ChangeLists. The process of building ChangeLists using Mocha is split into 3 phases:

1. **Discovery (Section 3.2.1).** To discover coherent slices that explain the shift inconsistency $\sigma_\ell$, we adapt the Domino (Eyuboglu et al., 2022) *slice discovery* method to our model comparison

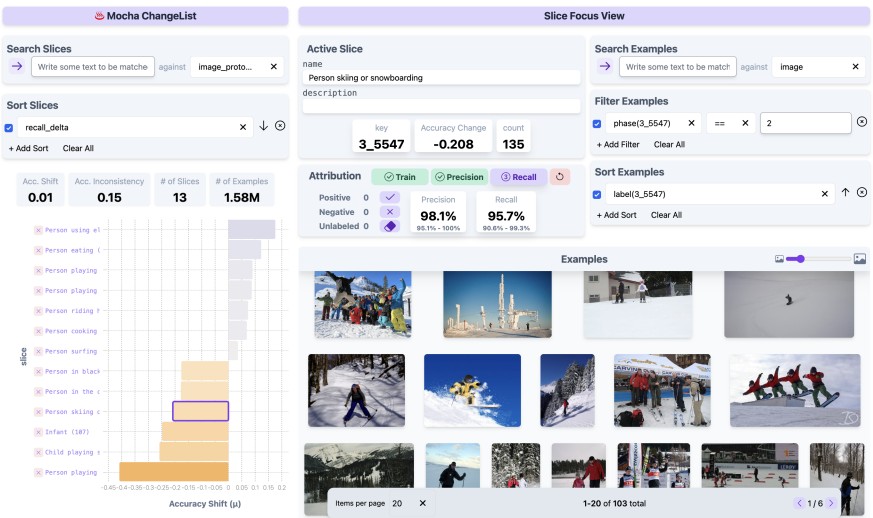

Figure 3: **A `ChangeList` produced using `Mocha`. (a)** *ChangeList View* shows data slices where model performance has improved and degraded. Users can navigate the slices either by issuing search queries or by sorting on size and performance shift. **(b)** *Slice Focus View* shows examples in the currently selected slice. During the attribution of discovered slices, it also provides rapid labeling tools needed for labeling importance weighted samples (see Section 3.2.2). The implementation of the `Mocha` GUI and back-end is written in Python using a common data-wrangling library.

setting. Additionally, we use Domino to generate text descriptions for each discovered slice $S^{(j)}$, which serve as initial attributions $A^{(j)}$. `Mocha` users can manually define slices using interactive tools for search, filtering and labeling, as well as add slices generated by any methods or sources.

2. **Attribution (Section 3.2.2).** Once slices are discovered, these initial slice attributions can be updated using interactive slice inspection in `Mocha`. A key problem is estimating the alignment of a slice $S$ with its attribution $A$, while collecting attribute realizations $\{a_i\}_{i=1}^n$ on a few informative examples labeled by the user. `Mocha` proposes *micro-labeling*: an importance sampling procedure driven by CLIP to find the most relevant examples for estimating alignment, coupled with an interface to rapidly label their attribution realizations. Slices with poor attribution alignment can also be updated to improve alignment using a fast training procedure.

3. **Release (Section 3.2.3).** Finally, once a `ChangeList` is finalized, it can be released as an *interactive* web application presented in the `Mocha` interface. Users can search and sort the `ChangeList` by the attributions and change metrics in order to read it quickly, and can continue to edit the `ChangeList` using `Mocha` at any time. `Mocha` also provides semantic text search over slices, using CLIP to find slices with similar image prototype or text attribution embeddings.

### 3.2.1 DISCOVERY

Slices are often sourced from metadata or extracted programmatically from the inputs (Goel et al., 2021b). When working with complex data types (*e.g.* images), many important slices are not annotated in metadata and cannot easily be extracted programmatically. The limited slices available are insufficient to explain the shift inconsistency, and we must turn to *slice discovery*.

Slice discovery for model comparison is the task of mining unstructured inputs $X$ for coherent slices that explain the shift inconsistency $\sigma_\ell$. Recent work has explored slice discovery frameworks for error analysis on *a single model* (Eyuboglu et al., 2022; Singla et al., 2021; Sohoni et al., 2020; d'Eon et al., 2022; Yeh et al., 2020; Kim et al.). Below we adapt the *Domino* framework (Eyuboglu et al., 2022) to our new task of explaining model differences in terms of unknown, unlabeled slices.

*Domino* takes as input trained models $v^{[1]}, v^{[2]} : \mathcal{X} \to \mathcal{Y}$ and a labeled dataset $\mathcal{D} = \{(x_i, y_i)\}_{i=1}^n \sim \mathcal{P}(X, Y)$, and outputs slicing functions $\Psi = \{\psi^{(j)} : \mathcal{X} \times \mathcal{Y} \to [0, 1]\}_{j=1}^k$ that partition the data into $k$ slices $\hat{\mathbf{S}} := \Psi(X, Y) \in [0, 1]^k$. *Domino* proceeds in 3 steps: (1) *embed* the dataset, (2) *slice* the resulting representation space, and (3) *describe* the discovered slices with natural language.

**Embed.** We embed the dataset $\mathcal{D}$ using an encoder $g_{\text{input}} : \mathcal{X} \to \mathcal{Z}$ ($\mathcal{Z} \in \mathbb{R}^d$), which yields embeddings $Z = \{z_i := g_{\text{input}}(x_i)\}_{i=1}^n$ for each example. Following Eyuboglu et al. (2022), we use CLIP (Radford et al., 2021), a cross-modal foundation model, as our encoder.

**Slice.** We discover slices by fitting a $k$-component mixture model to the embeddings $Z$ and model losses $\ell^{[1]}, \ell^{[2]}$. For mixture $S^{(j)}$, we assume $Z|S^{(j)}$ varies as multivariate Normal with diagonal covariance. The distribution of losses depends on the loss function $\ell$. In the zero-one loss case, we assume $\ell_{01}^{[1]}|S^{(j)}, \ell_{01}^{[2]}|S^{(j)}$ vary as categoricals. We then optimize the log-likelihood with expectation maximization. Like *Domino*, we use a hyperparameter $\gamma$ to balance the contribution of the embeddings and losses to the log-likelihood – higher $\gamma$ trades-off coherence for explanatory power.

**Describe.** Finally, to help users interpret discovered slices, we describe slices in natural language. We source candidate natural language phrases using a large, generative language model. We then identify descriptions $a^{(j)}$ which are closest in embedding space to the centroid of the each slice $S^{(j)}$.

For details on *Domino*, we refer the reader to Eyuboglu et al. (2022). Once the discovery phase is complete, the `Mocha` interface (Fig. 3) displays all discovered and user-specified slices (see Appendix A.4.1 for manual slicing), along with identified attributions and change metrics. Users can continue to perform discovery at any time in order to add additional slices to the `ChangeList`.

### 3.2.2 ATTRIBUTION

The goal of the attribution phase is to help users verify and edit discovered slices, while communicating the contents of each slice accurately. `Mocha` enables users to interactively (1) *edit* machine attributions by inspecting examples, to align them with the slice; (2) *estimate* alignment between the slice and its attribution; (3) *update* problematic slices that are poorly aligned with their attributions.

**Edit Descriptions.** When users interpret discovered slices, they typically attribute succinct concepts to slices e.g. "subjects wearing sunglasses" if most images in a slice show a person wearing sunglasses. This attribution allows them to draw conclusions such as "the model update improved by x% accuracy on subjects wearing sunglasses". `Mocha` provides interactive components (Fig. 3; discussed in Section 3.2.3) to quickly inspect slices in order to edit machine-generated descriptions. After editing, each slice has a single, textual attribution.

**Estimate Alignment.** Next, we want to determine the alignment of the slice $S$ with its attribution $A$ using precision $\mathrm{P}$ and recall $\mathrm{R}$. Measuring alignment lets users decide if a slice should be kept in the `ChangeList`, updated to improve alignment, or simply deleted. High precision implies that most slice examples satisfy the attribution i.e. $A = 1$, while with high recall, most examples that satisfy the attribution in the dataset are in the slice. Unfortunately, calculating $\mathrm{P}, \mathrm{R}$ requires exhaustively labeling the unknown attribute realizations $\{a_i\}_{i=1}^n$ for each example (*i.e.* labeling whether each example satisfies the slice, See 3.1), which is intractable to do for every slice.

Under a small labeling budget, we can only sample a few examples for labeling to estimate $\hat{\mathrm{P}}, \hat{\mathrm{R}}$. While we can estimate $\hat{\mathrm{P}}$ using simple random sampling (see Appendix A.2), naively estimating recall can have high variance (Owen, 2013; Kossen et al., 2021), since the number of false negatives (examples with $A = 1$ outside the slice) is frequently small relative to the dataset size $n$.

The key problem is how to construct a *proposal distribution* $q$ that upweights and samples "enough" false negatives to perform estimation via a procedure such as importance sampling (Owen, 2013). Our insight is to use CLIP (or any cross-modal foundation model) to construct one or more proposal distributions $q_i$, by ranking examples in terms of their similarity to the text attribution $A$. The advantage of using CLIP in this way is that it can provide an informative ordering of the examples in response to the wide range of (arbitrarily written) user attributions. A description of our estimation procedure is provided in Appendix A.2. Once the precision and recall are estimated, the user can apply a consistent decision rule (e.g. a minimum threshold on lower confidence bounds) in order to decide if the slice is satisfactory. If not satisfactory, the user can update it, which we discuss next.

**Update Slices.** For a slice $S = \psi(X, Y)$ with poor alignment with its attribution $A$, `Mocha` users can update the slice by training a new slicing function $\tilde{\psi}(X, Y)$, using logistic regression on CLIP embeddings (ignoring the label $Y$). Ideally, to improve alignment with $A$, $\tilde{\psi}$ should be aligned with $A$ on the dataset $\mathcal{D}$, i.e. $\tilde{\psi}$ is a good classifier of the attribution realizations $\{a_i\}_{i=1}^n$.

The main challenge is specifying labels for training $\hat{\psi}$, as the $\{a_i\}$ are either unknown, or partially known for previously labeled examples. We use a simple procedure to address this: use attribution labels if available (optionally with additional labeling), otherwise use the original slice labels $\{\psi(x_i, y_i)\}_{i=1}^n$. This updates $\hat{\psi}$ conservatively by matching $\psi$ where necessary, and allows `Mocha` to systematically improve the slice attribution alignment when a slice is updated. Note that for statistical validity we ensure no overlap between examples used for training or alignment estimation.

### 3.2.3 INTERACTION WITH MOCHA AND RELEASING CHANGELISTS

Finally, we provide an overview of how producers and users can interact with the `Mocha` interface (Fig. 3), and discuss the process of releasing `ChangeLists` in `Mocha`.

**Interaction.** `Mocha` contains several components to help users build and release `ChangeLists`,

1. **ChangeList View.** (Fig. 5) The left panel ① shows the current `ChangeList`. The `ChangeList` is displayed as a barplot against a chosen change metric, with the slice title and size annotated (①C). The user can select any slice for drilldown in the *Slice Focus View*. Users can sort slices in the `ChangeList` with change metrics (①B), or using text-based semantic search to order slices with the most similar image prototype or slice name embedding first (①A).

2. **Slice Focus View.** (Fig. 6) The right panel ② displays information about the slice selected in the `ChangeList`, including tools for performing attribution and navigating the slice. Slice summary statistics are shown along with the ability to edit its name and description (②A). The gallery (②F) enables quick inspection of slice examples, including example selection to display additional metadata. The gallery can be configured to view more or less examples at a glance, and can be sorted and filtered by slice, user and task labels, or any metadata (②D,E). They can also be sorted by semantic similarity to a text search, implemented using CLIP (②C). During attribution, the corresponding component (②B) guides the user through slice updates via training, and attribute quality estimation. The user first passes through an (optional) slice update where the slicing function is retrained using user provided labels. Then, for precision (and recall) estimation, the gallery displays the samples to be labeled for the estimation procedure of Section 3.2.2, and provides keyboard and mouse shortcuts for rapidly selecting and labeling examples. Once estimates are calculated, they are displayed in the same attribution component.

**Releasing `ChangeLists`.** After multiple rounds of discovery and attribution, a `ChangeList` can be released directly in the interactive `Mocha` web interface (Fig. 3). Users can navigate slices in released `ChangeLists` using the *ChangeList View*, and easily search for and drilldown into slices relevant to them. A key advantage of this method of release is that users can directly edit `ChangeLists` by performing additional rounds of discovery and attribution in `Mocha`.

## 4 DEMONSTRATING MOCHA ON REAL-WORLD API UPDATES

In this section, we discuss our takeaways from applying `Mocha` to three recent API updates and generating a `ChangeList` for each. First, we provide overview statistics summarizing the changes documented in the `ChangeLists`. Next, we dive into each API in detail, highlighting noteworthy changes, focusing on those where slice performance goes in the opposite direction as it does globally.

**Overview of `ChangeLists`.** `Mocha` enabled us to rapidly discover over 100 slices across three real API updates. For each discovered slice $S^{(j)}$, we compute the accuracy shift $\mu_{01}^{(j)} \approx \mathbb{E}[D]$ and test the null-hypothesis that the difference in accuracy $D$ is symmetric about zero using the Wilcoxon signed-rank test. On 103 slices, we find that at least one of the API's performance changed significantly (using the Bonferonni correction for multiple hypothesis testing, $\alpha = \frac{0.05}{k}$). Among these, 63 instances of an API's performance *degraded* significantly and on 52, the performance degraded by more than 5%-points. This phenomenon, where an update improves performance globally, but hurts performance on a coherent slice, has not been documented at this scale in the literature.

In Section 2, we motivate the need for `ChangeLists` by showing our updates introduced inconsistency that was not captured by the performance shift. We can quantify how much of the inconsistency is "explained" by our slices with the coefficient of determination $r^2$. Our `ChangeLists` achieve quite different $r^2$ on each update: 16.7% on EveryPixel, 12.4% on Google, and 5.3% on

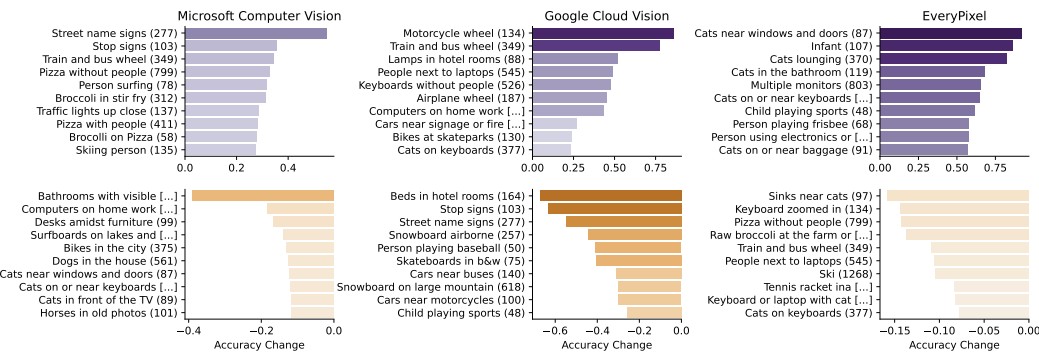

Figure 4: **Changes discovered by Mocha.** For each API update, we show the ten slices where the API performances increased *(top)* or decreased the most *(bottom)*. The $x$-axis shows the change in accuracy. The $y$-axis shows the name ascribed to the slice in attribution and its size in parentheses.

Microsoft. These low $r^2$ values highlight the difficulty of collecting a comprehensive set of slices and the importance of interactive ChangeLists that allow users to find additional slices.

The performance shift and inconsistency statistics above are defined in terms of a point-wise metric (*i.e.* accuracy), but we also compute precision, recall and F1. Of the 63 that showed statistically significant performance shifts, $74.38\%$ also saw changes in F1-score greater than $5\%$ and $61.1\%$ saw changes in F1 greater than $10\%$. Comparing precision and recall also provides insight into whether performance shifts are due to an increase in false positives or false negatives. Among the statistically significant performance shifts, $76\%$ exhibited a change in recall greater than $5\%$ and $10.3\%$ exhibited a change in precision greater than $5\%$. These statistics are summarized in Figure 2.

**ChangeList (Google Cloud Vision).** Google's API is used in diverse settings ranging from historic photos classification in newspaper archives (Greenfield, 2018) to managing visual assets in cloud storage (Kus, 2017). Even though the API improved *on average* after the update, it is important to identify fine-grained slices where performance has degraded. In Figure 4, we show 10 slices where performance degraded. Notably, the API's accuracy in detecting stop signs decreased by over $60\%$-points, a finding with potential safety implications. Post update, the accuracy of the API's "person" tag drops by $20.8\%$-points if the person is skiing or snowboarding. If they are playing baseball, accuracy drops by $40.9\%$-points, and if the photo is in black and white it is $17.7\%$-points. This last slice may be of particular interest to a newspaper using the API on archival photos.

**ChangeList (Microsoft Computer Vision).** Like Google, Microsoft's API is used in diverse settings and backs mobile applications and other intelligent software systems (Microsoft, b). Across the entire dataset, Microsoft's API improved significantly ($+4.0\%$ and $+5.9\%$ F1). However, our ChangeList includes 14 slices on which the performance drops significantly. For example, accuracy in tagging "horses" degrades by more than $10\%$-points in old, black and white photos.

**ChangeList (EveryPixel Image Recognition).** Unlike the other APIs, the average model performance degraded slightly between updates ($-0.5\%$ accuracy and $-2.9\%$ F1). Still, we were able to find data slices where the API improved. Notably, after the update, its "cat" detection improved across a broad set of contexts: near windows and doors, in the bathroom, and on or near keyboards. In contrast, the Microsoft API, which improved globally, exhibited significantly degraded performance on "cat" detection after the update.

## 5 CONCLUSION

MLaaS APIs are frequently updated with new versions, but providers rarely document how the API's behavior has changed. As a result, it is unclear to users how this affects downstream systems that depend on the API. In this work, we present Mocha, a framework for producing a ChangeList – a detailed report that highlights changes in performance using fine-grained slices of data. We hope that it is a step towards a future where providers include changelogs with every model update.

## ETHICAL CONSIDERATIONS

We highlight limitations with our proposed framework, which should be considered before use. These limitations also present potential avenues for future work. Our approach relies heavily on general-purpose pretrained models. In specialized domains (*e.g.* medicine, law), such models may not be readily available, limiting the applicability of our work. Furthermore, these models commonly exhibit social biases. For example, CLIP has been shown to encode harmful stereotypes in its representations Agarwal et al. (2021). Because we use CLIP embeddings to both identify slices and generate natural language descriptions, it is possible that a `ChangeList` may include offensive and harmful descriptions or groupings of data. As a result, it is critical that each `ChangeList` be carefully reviewed prior to release. There is also risk that a `ChangeList` may fail to document an important change in model behavior, potentially giving users a false sense of security. Future work should explore techniques for measuring the completeness of a `ChangeList`.

## REPRODUCIBILITY STATEMENT

We plan to release and open-source our implementation of `Mocha`, including the `ChangeLists` that we discussed in Section 4, as well as all metadata associated with the generation of these `ChangeLists` (outputs of slice discovery, labels collected during attribution, estimation of alignment metrics).

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

# A APPENDIX

## A.1 DISCOVERY

In this section, we provide details on the slice discovery techniques used in this work. In general, we follow closely the approach Eyuboglu et al. (2022).

We discover slices by fitting a $k$-component mixture model to the embeddings $Z$ and model losses $\ell^{[1]}, \ell^{[2]}$. For mixture $S^{(j)}$, we assume $Z|S^{(j)}$ varies as multivariate Normal with diagonal covariance. The distribution of losses depends on the loss function $\ell$. In the zero-one loss case, we assume $\ell_{01}^{[1]}|S^{(j)}, \ell_{01}^{[2]}|S^{(j)}$ vary as categoricals. The log-likelihood over the validation dataset is given as follows and mazimied using expectation-maximization:

$$\ell = \sum_{i=1}^{n} \log \sum_{j=1}^{\bar{k}} P(S^{(j)}=1)P(Z=z_i|S^{(j)}=1)P(\ell^{[1]}=y_i|S^{(j)}=1)^{\gamma}P(\ell^{[2]}|S^{(j)}=1)^{\gamma}, \quad (3)$$

Like *Domino*, we use a hyperparameter $\gamma$ to balance the contribution of the embeddings and losses to the log-likelihood – higher $\gamma$ trades-off coherence for explanatory power. A slice is *coherent* if the examples in it share something common. A set of slices have explanatory power if membership in those slices can explain the shift inconsistency.

## A.2 ATTRIBUTION: ALIGNMENT ESTIMATION

We would like to estimate the precision and recall with only a small amount of labeling effort:

$$P = \frac{\sum_{i=1}^{n} s_i a_i}{\sum_{i=1}^{n} s_i} \qquad R = \frac{\sum_{i=1}^{n} s_i a_i}{\sum_{i=1}^{n} a_i}$$

Consider one of the slices $S$ that was discovered in the first phase of `Mocha` (Section 3.2.1). Because $S = \psi(X, Y)$, we can compute the realizations of the slice variable $\{s_i = \psi(x_i, y_i)\}_{i=1}^{n}$ across our full dataset. On the other hand, we cannot access any of the realizations of the attributions $\{a_i\}$, since they are unknown.

**Estimating Precision.** We estimate precision directly using the standard approach of Monte Carlo estimation with simple random sampling (SRS). To estimate precision $\hat{P}$, we first sample $n_P$ examples to label with their attribution realizations $\{a_i\}$, and then compute the estimator $\hat{P} = \frac{\sum_{i=1}^{n_P} 1[a_i=1]}{n_P}$. We use a standard bootstrap procedure to compute a confidence interval around the estimated precision (Efron & Tibshirani, 1994).

**Estimating Recall.** Unfortunately, estimating recall efficiently is difficult since the number of false negatives (i.e. examples with $A = 1$ that lie outside the slice) can be small relative to the size of the dataset, making SRS an inefficient method with high variance in this setting. Beyond SRS, there are many approaches to sampling and estimation with a small sample size including stratified sampling (Parsons, 2014), importance sampling (Owen, 2013), ranked set sampling (McIntyre, 1952) and others, as well as adaptive variants (Bugallo et al., 2017). Estimating recall with limited labels has also recently received more attention in the machine learning community, particularly with adaptive approaches (Kossen et al., 2021; Marchant & Rubinstein, 2021; Poms et al., 2021).

Among these approaches, importance sampling is a strong and reliable baseline, and we leave the exploration of adaptive methods to future work. For simplicity, we reduce recall estimation to a two step process: (1) using *mixture* importance sampling (Owen, 2013) to estimate the proportion Q of examples with the attribution $A = 1$ in the complement of the slice; and (2) using a plug-in estimator for recall with the estimates for precision $\hat{P}$ and proportion $\hat{Q}$. Formally, we define Q,

$$Q = \frac{\sum_{i=1}^{n}(1-s_i)a_i}{\sum_{i=1}^{n}(1-s_i)}$$

In the first step, we use mixture importance sampling i.e. a simple variant of the Horvitz–Thompson estimator that is unbiased (Owen, 2013). Key to this method is the choice of the proposal distributions $q_i$, which upweight samples that are likely to be useful for estimation (Owen, 2013). Indeed,

the key problem is how to construct *proposal distributions* that sample "enough" false negatives for estimation via importance sampling (Owen, 2013). Good choices for the $q_i$ (i.e. those that lead to low variance estimates) would put higher weight on the less prevalent samples with $A = 1$, and lower weight on those with $A = 0$. At first glance, this appears impossible without labeling the attribute realizations $\{a_i\}$. While prior work has studied the estimation of classifier recall with limited labels (Kossen et al., 2021; Marchant & Rubinstein, 2021; Poms et al., 2021), these all reuse the classifier being evaluated to construct a proposal distribution. We do not have a classifier for arbitrary (user constructed) text attributions $A$ in our setting.

Instead, we propose a procedure that relies on a flexible method to construct proposal distributions. Our insight is to use CLIP (or any cross-modal foundation model) to construct one or more proposal distributions $q_i$, by ranking examples in terms of their similarity to the text attribution $A$. The advantage of using CLIP in this way is that it can provide an informative ordering of the examples in response to the wide range of (arbitrarily written) user attributions. This in turn leads to proposal distributions that are more likely to appropriately upweight samples that correspond to the concept $A$, which may have been arbitrarily selected by the user.

In detail, each example $x_j$ in the population is assigned a score $\lambda_{ij}$ based on inner-product search with respect to text queries $i \in [d]$ written by the user. Here, the user will write text queries that they think align with the attribution $A$. The similarity score $\lambda_{ij}$ serves as a useful proxy for whether the example satisfies the attribution $A$, and the ranking of examples by $\lambda_{ij}$ should correlate with the attribution realizations. Then, we construct a proposal distribution $q_i$ from each set of scores by first min-max scaling the scores, and then powering them in order to skew the distribution i.e. $q_i(x_j) \propto \left( \frac{(\lambda_{ij} - \min_k \lambda_{ik})}{(\max_k \lambda_{ik} - \min_k \lambda_{ik})} \right)^r$ for an exponent $r$. This serves to create a proposal distribution that assigns very low probability to examples that have the lowest scores.

Once the proposal distributions $q_i$ are created, we construct a mixture distribution $q_\alpha = \sum_i \alpha_i q_i$ with $\sum_i \alpha_i = 1$ (by default, we use the uniform mixture $\alpha_i = \frac{1}{d}$). We sample $n_Q$ examples from the mixture distribution with corresponding weights $w_j$ (with $w_j = \sum_i \alpha_i w_{ij}$) and user provided attribution realizations $a_j$. We can then estimate the proportion of examples $\hat{Q}$ in the slice complement with $A = 1$, as well as the recall $\hat{R}$ using an (unbiased) plug-in estimator,

$$
\hat{Q} = \frac{\sum_{j=1}^{n_Q} \frac{1[a_j=1]}{w_j} \cdot \frac{1}{n - \sum_{i=1}^n s_i}}{n_Q}, \qquad\qquad \hat{R} = \frac{(n - n_s) \cdot \hat{P}}{(n - n_s) \cdot \hat{P} + n \cdot \hat{Q}}
$$

We provide confidence intervals for recall by running a standard bootstrap procedure independently for both $\hat{P}$ and $\hat{Q}$, and combine these independent estimates to get bootstrapped estimates for recall. We then output the appropriate quantiles corresponding to the required confidence level.

Once the precision and recall are estimated, the user can apply a consistent decision rule (e.g. a minimum threshold on lower confidence bounds) in order to decide if the slice is satisfactory. If not satisfactory, the user can update it, as discussed in Section 3.2.2.

## A.3    Extended Description of the Longitudinal Database of API Predictions

### A.3.1    Image Recognition

**Task (Image Tagging).** In image tagging, the input $X$ is an image and category (*e.g. horse*) pair and the target $Y \in \{0, 1\}$ is a binary label indicating whether an object in the category is in the image. We consider the point-wise zero-one loss $\ell(y, \hat{y}^{[i]}) = \mathbf{1}[y = \hat{y}^{[i]}]$. We also report other metrics that are not point-wise: recall, precision, and F1-score.

**Dataset.** We use the Large Vocabulary Instance Segmentation (LVIS) dataset, a relabeling of the original Common Objects in Context (COCO) images (Gupta et al., 2019; Lin et al., 2014). The dataset has $n = 1,577,603$ examples. LVIS labels have two advantages over the original COCO labels. (1) LVIS includes over $1,203$ categories (compared to the $80$ in COCO), which better reflects the breadth of categories output by modern image tagging APIs. (2) LVIS provides *negative sets*, a set of images for each category where **no** instance of the category appears. This allows us to measure both the precision and F1-score of the APIs, while still using a long-tail set of categories.

Table 1: Estimates of precision and recall for measuring attribution alignment with slices found across all 3 API updates. Slices such as "stop signs" and "skateboard wheel" have low recall, so they may be rejected for inclusion in a final `ChangeList`, while all other slices have both precision and recall above 0.7.

| name | count | recall | precision |
| --- | --- | --- | --- |
| Snowboard airborne | 257 | 0.971 (0.94, 1.00) | 0.995 (0.98, 1.00) |
| Stop signs | 103 | 0.631 (0.54, 0.76) | 0.947 (0.89, 0.99) |
| Street name signs | 277 | 0.789 (0.73, 0.85) | 0.986 (0.97, 1.00) |
| Cats in the bathroom | 119 | 0.889 (0.83, 0.95) | 1.000 (1.00, 1.00) |
| Dogs in the house | 561 | 0.801 (0.75, 0.85) | 0.980 (0.97, 0.99) |
| Horses in old photos | 101 | 0.974 (0.92, 1.00) | 1.000 (1.00, 1.00) |
| Horses in rural settings | 446 | 0.913 (0.87, 0.96) | 0.930 (0.91, 0.96) |
| Surfboards on lakes and rivers | 35 | 1.000 (1.00, 1.00) | 1.000 (1.00, 1.00) |
| Surfboards in the ocean | 614 | 0.859 (0.82, 0.92) | 1.000 (1.00, 1.00) |
| Surfboards away from water | 173 | 0.959 (0.89, 1.00) | 1.000 (1.00, 1.00) |
| Motorcycle wheel | 134 | 0.795 (0.73, 0.86) | 1.000 (1.00, 1.00) |
| Train and bus wheel | 349 | 0.936 (0.89, 0.98) | 1.000 (1.00, 1.00) |
| Airplane wheel | 187 | 0.981 (0.96, 1.00) | 1.000 (1.00, 1.00) |
| Skateboard wheel | 186 | 0.663 (0.62, 0.72) | 1.000 (1.00, 1.00) |
| Skiing person | 135 | 0.957 (0.91, 0.99) | 0.981 (0.95, 1.00) |

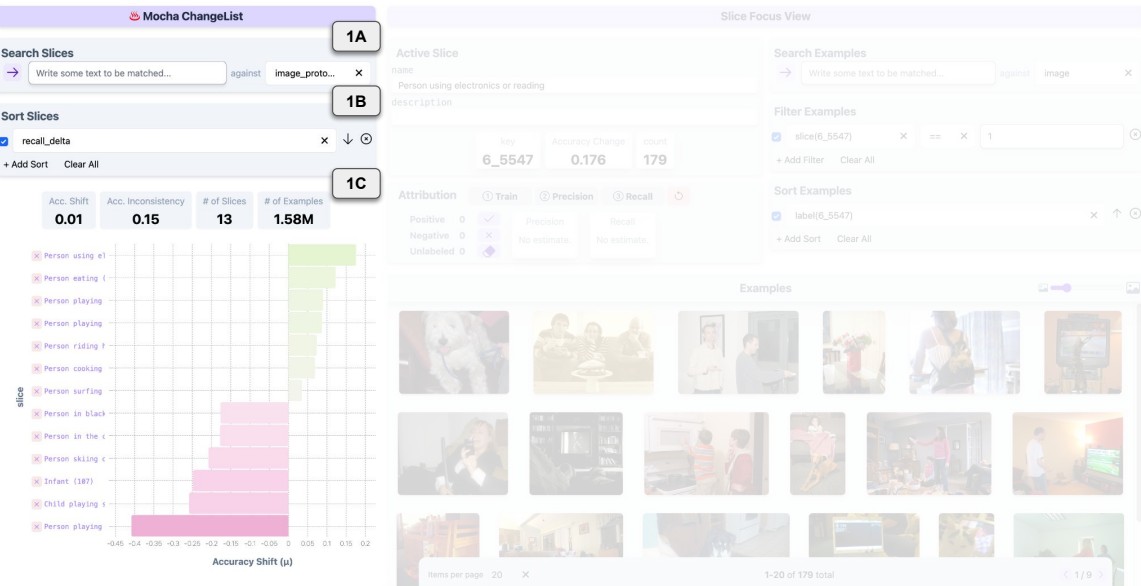

Figure 5: **Mocha ChangeList View**. The `Mocha ChangeList` panel consolidates information about the current `ChangeList`. Users can search for slices using text-based semantic queries, which match slices with the most similar image prototype or slice name (component 1A). Slices can also be ordered by associated metadata, such as change in performance or number of examples in the slice (component 1B). The barplot summarizes changes in a user-selected metric across the different slices (component 1C).

**APIs.** We consider three object detection APIs: Google's AutoML Vision Object Detection API (Google), EveryPixel's Image Keywording Service EveryPixel, and Microsoft Computer Vision Image Understanding API (Microsoft, a). The predictions are sourced from *History of APIs* (HAPI), a longitudinal database of API predictions (che). We use the raw outputs of the APIs and perform our own preprocessing that maps the labels output by the APIs to those in LVIS (see Section A.4 for details).

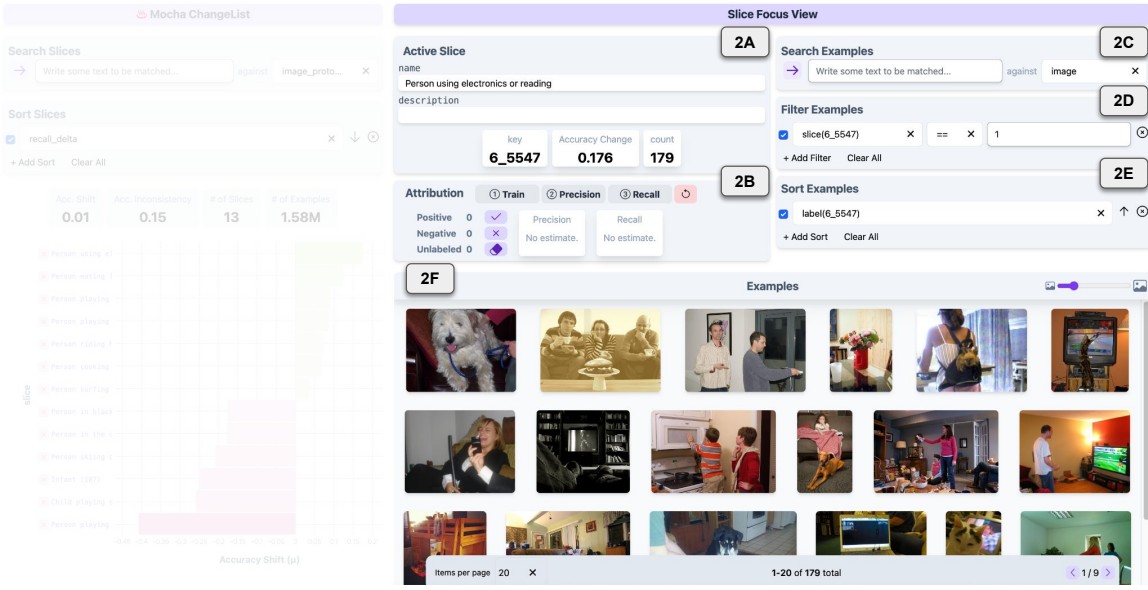

Figure 6: **Slice Focus View**. The slice focus vie enables granular inspection of different examples in the slice. Users can visualize characteristics of the selected (active) slice (component 2A) and manually label different attributes in the dataset (component 2B). Users can also search for examples that match unstructured text queries (component 2C) and filter and sort examples by existing or generated metadata (components 2D,2E). All examples are ordered in the gallery, which enables efficient data scrubbing.

**Reconciling labels.** The label set output by image recognition APIs will not necessarily match that of the evaluation dataset. For example, LVIS includes labels for 1,723 different object categories, while the 2020 version of the Google API output over 7,462 different object categories Gupta et al. (2019). In order to evaluate an API's performance on a dataset, we must first reconcile the two category sets. If an API outputs a category not in the LVIS vocabulary (*e.g.* "toboggan"), we want to map it to a more general category in the LVIS vocabulary (*e.g.* "sled"). To do so, we leverage the WordNet lexical database Fellbaum (1998), collecting for each category in LVIS all words with a more specific meaning (*i.e.* its hyponyms). We find the hyponyms of a category using the following procedure:

1. For each **category** in evaluation dataset, get the corresponding WordNet **synsets**. (LVIS categories are already based on WordNet synsets.)

2. For each **synset** compute all (direct and indirect) **hyponyms**.

3. For each **hyponym** collect all of its **lemmas** and filter them to down only include those whose most common noun word sense (based on WordNet sense ordering, see Daniel Jurafsky (2021)) is the hyponym. This gives us a mapping from each lemma to a list of categories.

   - *Note*: This serves to filter out improbable mappings. For example, the synset "mouse.n.02" is defined as "a person who is quiet or timid" and is a hyponym of "person". However, this leads to an odd mapping: a prediction of "mouse" is mapped to the category "person". To avoid this, we only consider the most common noun word sense of a lemma, which for "mouse" describes a rodent.

4. For each **lemma** select the category with the highest path similarity between its synset and the lemma's hyponym. This gives us a mapping from each lemma to a single category.

Using the resulting mapping, we can then "translate" the set of categories predicted by the API to the set of categories used in the dataset. If a predicted category is not in the mapping, we ignore it. This means that each predicted category is mapped to *at most one* dataset category. If the object

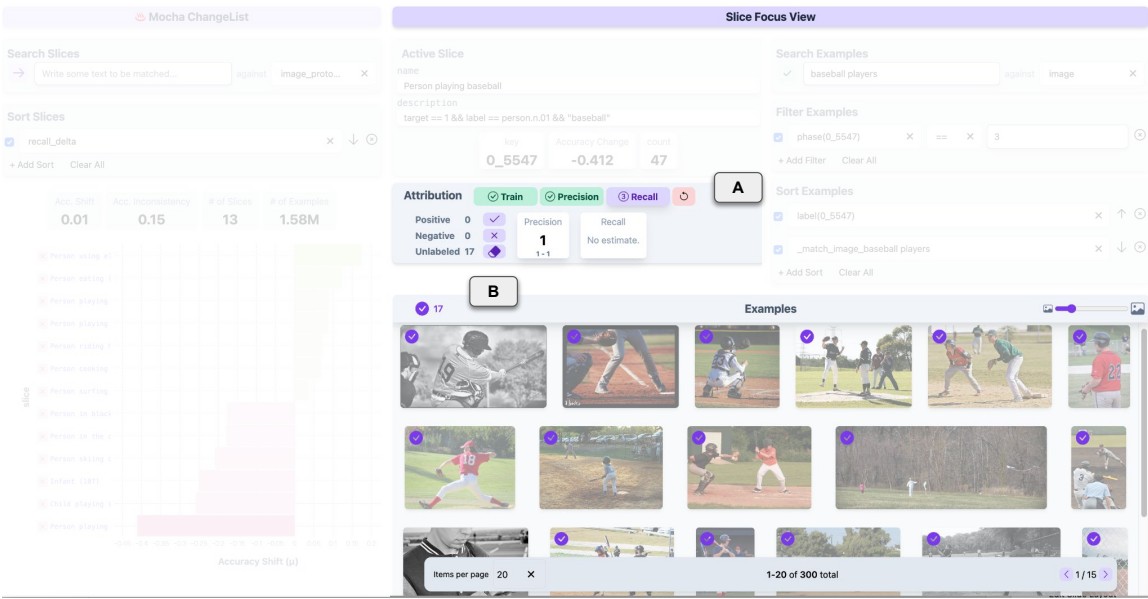

Figure 7: **Slice Refinement and Coherence Statistics**. The attribution panel (A) provides an fast zero-one data labeling interface, which allows users to efficiently refine slices, bootstrap refiner models, and compute coherence metrics (e.g. precision, recall) on the chosen slice. Users batch select examples in the gallery (B) and select one of three options to label: 1) positive, 2) negative, 3) unlabeled (i.e. erase).

belongs in both specific and more general categories (*e.g.* "canoe" and "boat") the API is expected to output both the specific and general categories. This is based on the recommended evaluation approach provided by the LVIS authors Gupta et al. (2019).

## A.4 EXTENDED DESCRIPTION OF MOCHA

### A.4.1 MANUALLY GATHERING SLICES

Manually gathering slices is a critical process for refining outputs of slice discovery methods (SDMs) and for creating slices that were not automatically discovered. However, a manual step requires scalable data exploration, which is difficult to do with large datasets. In Mocha, users can rapidly scrub through data in the gallery and label examples to assign them to the appropriate slice (Fig. 7). Users can also create their own slices and label examples that are part of that slice.

Mocha also leverages image-text foundation models, like CLIP, to perform similarity search between image examples and text queries (Fig. 8). Similarity search can reduce the burden of having to scrub through large datasets when the attributes of interest are not labeled. Similarity searches can also be used to find semantically meaningful groups of images, which can expedite manual slice discovery workflows.

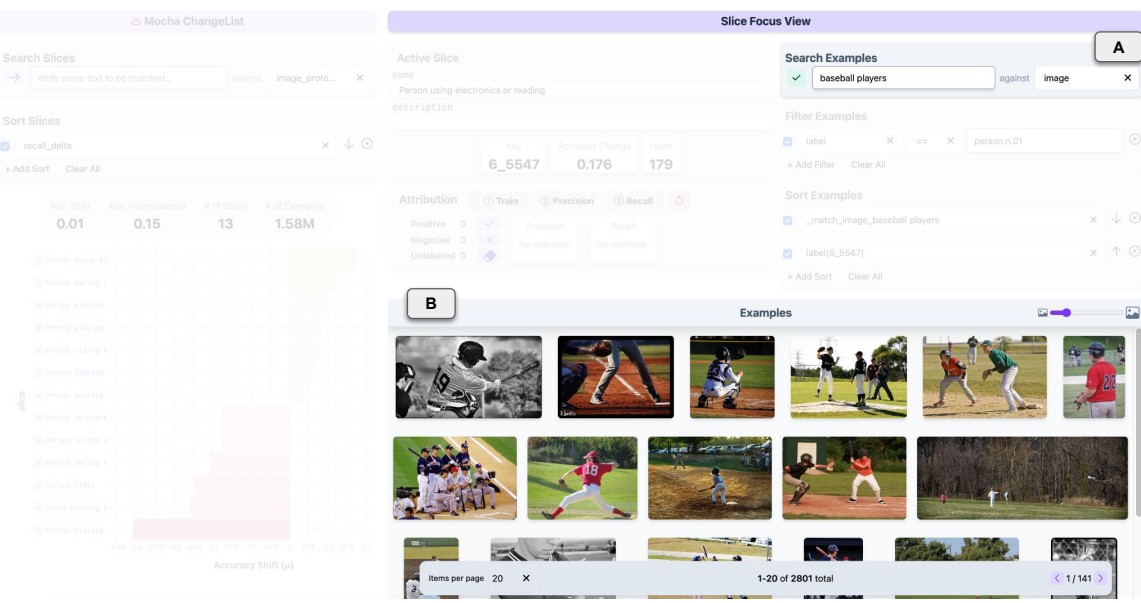

Figure 8: **Searching for Examples**. `Mocha` supports semantic similarity search between images and unstructured text using image-text foundation models, like CLIP. Based on the user search query (A), images in the selected slice (or, if no slice is selected, entire dataset) are sorted by their similarity to the query (B).

