# OpenReview forum: "Model ChangeLists: Characterizing Changes in ML Prediction APIs"
_ICLR.cc/2023/Conference — Submitted to ICLR 2023_

### Official Review · Reviewer_Q1WT · 2022-10-24

**Confidence:** 4
**Correctness:** 3
**Technical Novelty And Significance:** 1
**Empirical Novelty And Significance:** 1
**Recommendation:** 3

**Clarity, Quality, Novelty And Reproducibility:**

Clarity:  This paper is super clear, with good writing and illustrations.

Quality:  The quality of the presentation is quite good.

Novelty:  I think this work has minimal novelty.

Reproducibility:  If the code (and data!) is released as promised, it should be generally reproducible.

**Strength And Weaknesses:**

Strengths:
+ Blackbox understanding of ML models is an interesting and growing problem
+ The Domino method this builds on is interesting

Weaknesses:
- Very low novelty
- No user studies to validate Mocha’s effectiveness
- No comparison with existing ML explanation tools
- Insufficient comparison of design tradeoffs
- Even ignoring all the problems with this paper, I can’t believe that it serves a purpose in its current form.  Basically the granularity of the provided information just seems wrong.  The authors are able to cherry pick one useful example (black and white imagery classification got worse), but in general the slice labelings seem all over the place.  In practice, a real user of these APIs likely has labeled images they care about, and would generally prefer to measure the ML system’s performance on their task directly.


**Summary Of The Paper:**

This paper proposes a user interface for analyzing differences between ML models (ostensibly public cloud models, but it could be anything).  While the idea of segmenting model input for the purpose of model understanding is interesting, this paper is basically describing a fairly primitive GUI for the (already published, Eyuboglu et al 2022) method that does the heavy lifting.  So most of the core novelty has already been described.

Quite frankly, this paper comes across as a UI paper written by a ML person, but doesn’t score well in either category.  I can’t see its relevance to ICLR.  This paper might be a better fit for a visualization conference, but even there, the concepts behind these visualizations seem relatively standard, so it is hard for me to see what contribution is being made to the visualization literature.

**Summary Of The Review:**

I think the paper has low novelty and is a bad fit for ICLR.

---

### Official Review · Reviewer_NHrB · 2022-10-30

**Confidence:** 3
**Correctness:** 4
**Technical Novelty And Significance:** 2
**Empirical Novelty And Significance:** 3
**Recommendation:** 5

**Clarity, Quality, Novelty And Reproducibility:**

Paper is very well written and easy to follow. There are plans to release the code and without it is it hard to reproduce.

**Strength And Weaknesses:**

Very well motivated and useful paper, with sound methodology, and could have impact in practice.

**Summary Of The Paper:**

This paper provides a tool for understanding the performance changes between different updates of ML as service applications. The developed tool is interactive and helps the users discover meaningful coherent different data slices where the performance is changing on, called ChangeLists. The model is built on CLIP and Domino, and provides a web based interface that could be very helpful for users of MLaaS to figure out if updates should be adapted based on their specific application.

**Summary Of The Review:**

While I agree with the paper contribution being significant, I think this is more a demo paper and/or visualization paper. The novelty on the methodology side is limited and it is more a clever usage of prior methods/models (Domino/Clip). I am happy if the paper is accepted since it probably will be useful but I am not sure if it passes the novelty bar of ICLR.

---

### Official Review · Reviewer_LdES · 2022-11-03

**Confidence:** 4
**Correctness:** 4
**Technical Novelty And Significance:** 4
**Empirical Novelty And Significance:** 3
**Recommendation:** 6

**Clarity, Quality, Novelty And Reproducibility:**

I think the paper is in general well-written and solves a novel problem. I have some concerns with regards to the clarity of presentation which has been mentioned in the weaknesses section of this review.

I do not have any reproducibility concerns contingent on the fact that my above concerns in the weaknesses section are adequately addressed.



**Details Of Ethics Concerns:**

The paper mentions ethical concerns which stems from the fact that pre-trained models are used which are known to exhibit social biases. However, this is not a concern of the method since this can be mitigated by using pertained models that learn better and fair representations.

**Strength And Weaknesses:**

Strengths:

1. A very interesting problem with practical implications.
2. Well-written paper and easy to follow modulo some aspects which are mentioned in weaknesses.
3. Experiments on different publicly available deep learning APIs show the usefulness of the proposed approach.

Weaknesses:
1. I think the description of attribute text isn't very clear. What are the instance level attributions a_i? Are these the generated descriptions from a generative model which were deemed closest in embedding space to the centroids (as described in Section 3.2.1)? The paper mentions that users can modify this attribution. Do we have a single text attribution for each slice or multiple? I think adding some examples of attributions more specifically where it is described

2. It would help to write out the loss used in discovering the slices using EM instead of just referring to Domino (perhaps in the appendix). For instance this line is unclear "Like Domino, we use a hyperparameter γ to balance the contribution of the embeddings and losses to the log-likelihood – higher γ trades-off coherence for explanatory power." What does explanatory power mean? what does coherence mean?

3. Similarly in the last paragraph on page 7, "For a slice S = ψ(X, Y ) with poor alignment with its attribution A, Mocha users can update the slice by training a new slicing function ψ ̃(X, Y ), using logistic regression on CLIP embeddings.". It is not clear what does this precisely mean. CLIP embeddings encode image X to some latent space Z, how does Y play a role? does the logistic regression classifier take as input both Z and Y? I think these details are important in fully understanding the implementation in this paper.

4. The estimation of Recall for attribution and slice alignment using importance sampling isn't very clear. The paper proposes to use a mixture distribution with one q_i per user generated text-description. My main confusion is this, does the user write multiple text annotations for each attribute or we have one annotation for an attribute? This goes back to my first point. Describing the attributes more clearly is necessary for better readability of the paper in my opinion.

5. The formula for \hat{Q} is not clear. The paper claims we do not have access to instance level attribute realizations for every image in the dataset. So, importance sampling is used to get n_Q samples and then manually annotate their attribute realizations. However, the formula seems to sum over all n samples in the dataset in the numerator. Is this a typo?

**Summary Of The Paper:**

This paper tackles an interesting problem of explaining changes between two different deep models in terms of detailed, fine-grained changes in performance across different subsets of data, called slices. This is done by first identifying distinct subsets of data that correspond to some coherent concept, these subsets are then automatically annotated with natural language descriptions which the users can edit/modify. The differences between the two models in then described in terms of "Changelists" which is a collections of slices of data along with annotations and performance metrics on these slices.



**Summary Of The Review:**

My final recommendation is a weak accept. I think the concerns regarding clarity should be addressed before acceptance.

---

> ### Author Response · Authors · 2022-11-19
> **Response to Reviewer LdES**
>
> We thank reviewer LdES for their thoughtful and thorough review. We address their concerns regarding the clarity of presentation below.
>
> 1. We agree the distinction between the slice and instance level descriptions could be better explained. We added additional explanation and a back-reference to the section where we defined $a_i$.
> 2. Due to space limitations in the main body, we’ve added the loss function and more details on these terms to the appendix.
> 3. The logistic regression classifier takes as input only $Z$ (ignoring $Y$). We’ve updated the text to explain this more clearly.
> 4. We added the following text to clarify: “After editing the machine-generated descriptions, each slice has a single, textual attribution.”
> 5. We thank the reviewer for pointing out this typo. The range of the sum goes over $n_Q$ examples.

---

### Official Review · Reviewer_C4kp · 2022-11-03

**Confidence:** 3
**Clarity, Quality, Novelty And Reproducibility:** Paper is written pretty well. Novelty…
**Correctness:** 3
**Technical Novelty And Significance:** 2
**Empirical Novelty And Significance:** 2
**Recommendation:** 3

**Strength And Weaknesses:**

**Strengths**

 1) The paper addresses an important problem of trying to understand how model updates impact different data slices. In particular, the issue of global improvement in model performance at the risk of lowered performance in certain subgroups is very important to know when deploying production model changes.

2) Paper knits all components together in an interactive framework and shows results using production models from Microsoft and Google.

**Weakness**

1) I believe one of the paper's biggest flaws is the lack of technical novelty. While I do acknowledge that the overall framework drives value, the individual components leverage existing prior art such as Domino (Eyuboglu et al. 2022), Mixture models and importance sampling procedures making incremental technical improvements. I am also not quite sure if this paper fits in within the core technical track of a conference like ICLR. It might be a very good fit for UI, demo or Industry track.

2) I also do believe that some user studies are also important to assess how user friendly Mocha is. Overall the paper does present this as a framework which can potentially overwhelm the user due to its multiple technical intricacies.  Ideally, Mocha should be useful for data scientists and business intelligence teams also and not just be applicable to core AI scientists to drive real value.

**Summary Of The Paper:**

The paper provides an interactive framework called Mocha for an end-user to view and edit change lists.  This notion of change lists are also one of the key contributions of this paper to explain performance shift and shift inconsistency by analyzing changes in fine grained data slices. The paper leverages techniques primarily motivated from Domino (Eyuboglu et al. 2022) and discovers the slices using a mixture model. Attributions are inferred using importance sampling driven by CLIP followed by alignment specific sanity checks. The paper presents empirical results comparing production models from Microsoft and Google to view these change lists.

**Summary Of The Review:**

Overall, I find this paper to be a good read in terms of providing a framework to drive value for end users. My major concerns are on (a) technical novelty of individual components and (b) user effectiveness studies to assess how viable is it to use Mocha.  Also, I feel this paper should be more well placed rather than being considered in core technical track of ICLR as it might have good value as a demo or industry track paper at ICLR or other similar top tier conferences.

---

### Author Response · Authors · 2022-11-19
**General Response to Reviewers**

We thank the reviewers for their thoughtful review of our manuscript. Below we summarize points made by multiple reviewers highlighting the strengths of our paper. Then, we respond to the two main concerns raised by reviewers. We've uploaded an updated version of the manuscript.

**Common Strengths**

*Important motivation and an interesting, novel problem.*

- “This paper … solves a novel problem.” - **LdES**
- “A very interesting problem with practical implications.” - **LdES**
- “Very well motivated.” - **NHrB**
- “An interesting and growing problem.” - **Q1WT**
- “The paper addresses an important problem.” - **C4kp**

*Contributions are of significant practical value.*

- “Overall, I find this paper to be a good read in terms of providing a framework to drive value for end users.” - **C4kp**
- “Experiments on different publicly available deep learning APIs show the usefulness of the proposed approach.” - **LdES**
- “Useful paper, with sound methodology, and could have impact in practice.” - **NHrB**

*Clear Presentation.*

- “Well-written paper and easy to follow.” - **LdES**
- “Paper is very well written and easy to follow.” - **NHrB**
- “Paper is written pretty well.” - **C4kp**
- “This paper is super clear, with good writing and illustrations. The quality of the presentation is quite good.” - **Q1WT**

***Common Concerns***

*Poor fit for ICLR due to limited methodological novelty.* Reviewers **C4kp** and **Q1WT** expressed concerns about the novelty in the methods used in our paper.

We focus on an understudied problem, propose a novel framework for addressing it, and instantiate that framework with simple, known methods. We use this framework to perform a first-of-its-kind analysis that highlights the practical importance of the problem to the machine learning community. As reviewer **NHrB** put it, ours is a “very well motivated and useful paper, with sound methodology, and could have impact in practice”.

Our decision to submit to ICLR (as opposed to visualization or HCI conferences, like some reviewers suggested) was motivated by the audience we hoped to reach with our findings: we believe that the machine learning community (*i.e.* the ones developing the models) should study how to communicate changes in models to users. This is especially important as MLaaS APIs are increasingly used by organizations and individuals who lack a background in machine learning and a knowledge of best practices in model validation.

*Missing user study.* Reviewers  **C4kp** and **Q1WT** comment that a user study would help in understanding how user-friendly the Mocha interface is. We validate Mocha by applying it to a set of real-world API updates, demonstrating how the framework can be used to discover and report fine-grained changes in performance. Reviewer **LdES** mentions this, saying “Experiments on different publicly available deep learning APIs show the usefulness of the proposed approach.” While we agree that a user study would provide further validation of the interface design, we believe the paper stands on its own without it. We hope to perform a user study in future work.

Reviewer **LdES** requested various additional clarifications, which we address in reviewer-specific comments.

---

### Decision · Program_Chairs · 2023-01-20

**Decision:**

Reject

**Justification For Why Not Higher Score:**

I am sympathetic to the fact that because this is a paper that introduces a new problem setting, it is perhaps not necessary (or even desirable!) for the paper to also introduce new methodology. However, the concerns about evaluation seem significant enough to me that I am unable to recommend acceptance at this time. Even without a user study, is there some way to make the case for model changelists more strongly?

**Justification For Why Not Lower Score:**

N/A

**Metareview: Summary, Strengths And Weaknesses:**

This paper presents the concept of model changelists, which describe the differences in performance between multiple versions of a model aimed at the same task. The key insight is to describe differences in the model's predictions in terms of data slices, a concept introduced in previous work.

Strengths:
* The idea of changelists seems to be a new and potentially interesting idea.

Weaknesses:

* Although the idea of changelists is novel, the paper does not seem to introduce new methods on the machine learning side or new principles on the visualization side.
* Several reviewers felt that more evaluation would be needed to demonstrate the potential effectiveness of changelists.

**Summary Of Ac-Reviewer Meeting:**

N/A